# The Spatial Spillover Effects of Fiscal Expenditures and Household Characteristics on Household Consumption Spending: Evidence from Taiwan

**Hao-Chen Huang** [1,*] **, Chen-Lin Yuan** [2] **and Ting-Hsiu Liao** [3]

1  Department of Public Finance and Taxation, National Kaohsiung University of Science and Technology, Taipei 10650, Taiwan
2  Business Intelligence School, National Kaohsiung University of Science and Technology, Yanchao District, Kaohsiung 82444, Taiwan
3  Graduate Institute of Tourism Management, National Kaohsiung University of Hospitality and Tourism, Xiaogang District, Kaohsiung City 812, Taiwan
*  Correspondence: haochen@nkust.edu.tw

**Abstract:** The main purpose of this study is to explore the determinants of average household consumption spending in counties and cities from the two aspects of government fiscal expenditure and household characteristics. A spatial econometric model, the spatial Durbin model, was used to analyze Taiwan's county-level and municipal panel data from 2000 to 2020. Global spatial autocorrelation and local spatial autocorrelation were applied to examine the overall degree of spatial agglomeration of average household consumption spending in Taiwan and the agglomeration status of specific counties and cities. The empirical results show that the average consumption spending per household of all counties and cities in Taiwan presents spatial autocorrelation, and the agglomeration of specific counties and cities is affected by different ruling parties of the central government. In terms of direct effects, the average consumption spending per household in local counties and cities is influenced by household characteristics, including average disposable income per household, average number of employees per household, and average living area per capita. In terms of the spatial spillover effect, the average consumption spending per household in local counties and cities is influenced by household characteristics of the neighboring counties and cities, including the average disposable income per household and the average living area per capita. Surprisingly, local economic development expenditure and local expenditure on education, science, and culture have no significant impact on the average consumption spending per household in counties and cities. The results of this study can be taken as a reference for government policymaking.

**Keywords:** fiscal expenditure; household characteristic; consumption spending; spatial autocorrelation; spatial Durbin model

## 1. Introduction

Household consumption is critical to stimulating economic growth. Consumption is the engine, source, and goal of social production and development. Moreover, it is a considerable driver and a significant and enduring contributor to economic growth. The study of consumer spending is a critical economic issue. The proportion of consumer spending in total expenditure exceeds about 90% of the GNI in developing countries but falls to about 60% in wealthy countries (Almosabbeh 2020). Many economists have studied household consumption spending as one of the key determinants of a nation's well-being (Duesenberry 1949; Friedman 1957; Keynes 1936). The long-term gap between urban and rural areas in Taiwan has resulted in a meager consumption rate in its counties and cities, except in Taipei, New Taipei, Taoyuan, Taichung, Tainan, and Kaohsiung. In addition, it has also resulted in a widening consumption gap between urban and rural areas, regions,



and different income groups. The two above factors have seriously hindered consumption growth in Taiwan and its contribution to economic and social development.

Regional difference has always been one of the most significant concerns of geographers, economists, and governments. Regional difference is commonplace in economic development, while residents' consumption difference directly reflects differences in economic and social development. However, in terms of the spatial dependence of household consumption spending, economic activities of household consumption in a specific geographical area do not exist independently; there is a spatial correlation with neighboring geographical areas. The distribution of household consumption spending or consumption behavior has specific spatial rules. The amount of household consumption spending in different geographical areas may be affected by local factors and neighborhood effects. However, an important premise for traditional econometrics is to assume that the study objects are independent of one another, which does not conform to the actual situation. Traditional econometric models assume that space objects are unrelated and homogeneous, and most adopt ordinary least squares (OLS) to estimate the regression model. Due to the neglect of spatial effect, the regression model generally had errors, leading to a lack of precision regarding estimated results and inferences drawn from the regression model (LeSage and Pace 2009). Traditional econometric models have certain limitations in spatial relations and model analysis, and it is difficult to determine factors that affect household consumption spending in counties and cities.

This study adopted the research method of spatial econometrics to explore the determinants that affect the average household consumption spending in counties and cities. A comprehensive review of previous literature shows that it is relatively rare to explore the topics of factors affecting household consumption spending from the perspective of spatial effect. This study analyzed fiscal expenditures and household characteristics from the standpoint of spatial effect. First, regarding fiscal expenditures, previous studies have demonstrated a significant relationship between government expenditures and private consumption spending (Bailey 1971; Bernardini and Peersman 2018; Bouakez and Rebei 2007; Evans and Karras 1996; Ho 2001; Samadi and Sayedi 2012). Whether government expenditures damage or stimulate private economic activities is a key issue in Taiwan's economy, especially for economic revitalization in counties and cities. Therefore, it is necessary to understand how fiscal expenditures from local governments affect private household consumption spending. This study focused on the impact of productive expenditures (economic development expenditures, and educational, scientific, and cultural expenditures) from government fiscal expenditures on average household consumption. Second, in terms of household characteristics, previous studies have demonstrated that different households have different economic conditions that affect household consumption spending (Lee and Mori 2019; Levay et al. 2021; Kozyreva et al. 2021; Tscharaktschiew and Hirte 2010). Therefore, we must understand how household characteristics affect household consumption spending. This study specifically focused on the impact of household characteristics on average household consumption spending, such as average disposable income per household, the average number of employed persons per household, the ratio of self-owned residences, the average living area per person, and the ratio of catering expenses to consumption spending. As previous literature seldom used spatial effect analysis, the purposes of this study are as follows: (1) to explore whether average household consumption spending in Taiwan's counties and cities has an overall agglomeration of spatial autocorrelation as well as the agglomeration in specific counties and cities, (2) to explore the direct effect of fiscal expenditures and household characteristics in local counties and cities on average household consumption spending in local counties and cities, and (3) to explore the spatial spillover effect of fiscal expenditures and household characteristics in neighboring counties and cities on average household consumption spending in local counties and cities.

In the next chapter, we will explore the literature and outline our hypothesis development. The following research methods illustrate data and samples, research variables,

and the spatial weight matrix. The empirical results include narrative statistics, correlation analysis, spatial autocorrelation verification, a LISA cluster map, the Hausman test, spatial Durbin model analysis, and the analysis of the direct and spatial spillover effect. Finally, we will present conclusions, policy implications, research limitations, and future research directions.

## 2. Literature Review

### 2.1. Application of the Spatial Econometric Model in Consumption Spending

Research on consumption spending is an important issue, but it has been mostly analyzed by traditional econometric models, such as the panel data model and even less by the spatial econometric model. A review of past studies using spatial econometric models runs as follows (Bao and Chen 2017; Filippini et al. 2009; Funashima and Ohtsuka 2019; Lv et al. 2019). For example, Funashima and Ohtsuka (2019) explored the spatial crowding-out and crowding-in effects of government expenditure on Japan's private sector. They argued that policymakers should consider spatial spillovers and regional differences, as well as boost regional economies by stimulating private demand. Lv et al. (2019) discussed the determinants of the impact of urbanization on energy consumption and whether the growth of energy consumption in one province in China has a demonstration effect and spillover effect on surrounding areas. Bao and Chen (2017) studied the influencing factors of water consumption efficiency in China and mentioned that the water consumption efficiency of different provinces has significant spatial autocorrelation characteristics. They further point out that the water consumption efficiency of a province may not only be affected by its socio-economic and ecological environmental indicators, but also by the efficiency of water use in its neighboring provinces. Filippini et al. (2009) explored the influencing factors of antibiotic consumption in Switzerland and report that personal income, population structure, doctor density, and drug price are all decisive factors affecting the per capita consumption of antibiotics in all regions.

There is limited research regarding the application of spatial econometrics on average household consumption spending. The reason is that spatial econometrics, developed to solve spatial autocorrelation problems, is a branch of econometrics in which there has been far less research than in traditional econometrics. In the past, most research had still applied traditional econometrics as the primary method. Presently, research on the application of spatial econometrics on consumption focuses on energy consumption expenditures or medical consumption expenditures. In contrast, research on household consumption spending is relatively rare. The contribution of this study is to research household consumption spending from the method of spatial econometrics to address the gap in previous research. Moreover, this study explored the factors that affect household consumption spending from fiscal expenditures and household characteristics, which is not only a topic unstudied in previous literature but also a contribution to future research.

### 2.2. Hypotheses Development

#### 2.2.1. Fiscal Expenditure and Consumption Spending per Household

Government expenditures are one of the main tools of macroeconomic stabilization policy. One heatedly debated issue in the macroeconomic literature is the impact of government expenditures on private consumption expenditures (Asimakopoulos et al. 2021; Özerkek and Çelik 2010). The Keynesian IS-LM model, for instance, argues that consumption increases as government expenditure increases. Conversely, the standard real business cycle (RBC) model argues that increased government expenditure should lead to a decline in consumption. Bailey (1971) first proved that there may be a certain degree of substitutability between government expenditure and private consumption. That paper established an effective consumption function, analyzed whether there is substitutability and complementarity between government expenditure and personal consumption spending, and held that there is complementarity between them. Barro (1981) examined the direct impact of government purchase of goods and services on consumption utility.

That study argued that a short-term increase in government expenditure boosts the increase in household consumption, but the increase of the latter is less than that of the former. The same applies to the long term, but at a lesser degree. Previous studies have demonstrated that there is a significant relationship between government expenditure and private consumption spending. Some studies report that there is a crowd-out relationship between fiscal expenditure and private expenditure (Bailey 1971; Bouakez and Rebei 2007; Ho 2001), while other studies have argued that there is a direct relationship between fiscal expenditure and private expenditure (Bernardini and Peersman 2018; Evans and Karras 1996; Samadi and Sayedi 2012). It is certain that government expenditure could influence private consumption activities through different channels (Funashima and Ohtsuka 2019).

Past scholars differed on whether government expenditure increases consumption spending. For example, Ho (2004) suggests that government spending in Japan leads to a decrease in private consumption spending. However, Karras (1994) has demonstrated in some countries that government fiscal expenditure improves the marginal utility of household consumption. Ho (2001) studied the relationship between government expenditure and household consumption in 24 OECD countries and found no significant relationship between them in a single country. However, when analyzing the data of multiple countries, Ho pointed out an obvious substitution relationship of government expenditure with household consumption.

In a study of the spillover effect, Kameda et al. (2021) mention that, because the local economy has strong interdependence without the border effect, it is easy for government expenditure in the local economy to spill over to other local economies. Therefore, this study established the following hypotheses, where H1a is the direct effect, and H1b is the spatial spillover effect.

**Hypothesis 1a.** *Fiscal expenditure (economic development expenditure, expenditure on education, science, and culture) of local counties and cities has a significant influence on the average consumption spending per household.*

**Hypothesis 1b.** *Fiscal expenditure of neighboring counties and cities (economic development expenditure, expenditure on education, science, and culture) has a significant influence on the average consumption spending per household in local counties and cities.*

### 2.2.2. Household Characteristics and Consumption Spending per Household

Becker (1965) explained the classical demand theory, in which consumption is regarded as the output of the household production function. A household is the basic economic unit of a society. However, the household structure varies from city to city, because the household is heterogeneous and differs in size and composition (Tscharaktschiew and Hirte 2010). In terms of urban–rural differences in household consumption, Kozyreva et al. (2021) pointed out that there are differences in urban–rural consumption inequality. For instance, in China the expenditure of urban households grows much faster than that of rural households. In terms of differences in household characteristics, Levay et al. (2021) have suggested that income and household size are the most important determinants of household consumption. Lee and Mori (2019) studied the impact of demographic characteristics on conspicuous consumption and noted that conspicuous consumption of residents living in high-status houses in Singapore increases by 25%. Past research has shown that household characteristics do impact household consumption spending.

As the economy develops, the more members in a family who are employed, the more household income they will earn. Gupta and Kishore (2022) point out that the unemployment of breadwinners leads to an immediate and significant decline in household consumption spending. The decline is steeper for urban households and for the lowest and highest deciles. Abundant disposable income for members of a household means they have more to consume, but exorbitant housing prices affect household consumption. The

size of the average living area per person (LAP) is related to housing prices. For example, Suari-Andreu (2021) and Carroll et al. (2011) found a strong correlation between housing price evolution and household consumption. Since housing prices in Taiwan have always been high, the cost of owning a spacious house is bound to be high. Excessively high housing prices will undoubtedly affect household consumption spending. Compared to other rural counties and cities, the average living area per person (LAP) of households in several large urban counties and cities in Taiwan, such as Taipei City, New Taipei City, or Taichung City, is much smaller than that in rural counties and cities, such as Changhua County, Chiayi County, and Pingtung County, with the same housing price. Household spending on purchases or rentals accounts for a high proportion of consumer spending. The desire for a larger average living area per person (LAP) may affect household spending. Typically, the neighborhoods surrounding urban counties and cities are rural. High household consumption spending in urban counties and cities causes many people to move to neighboring areas with relatively lower expenditures. The average living area per person (LAP) of households in these rural counties and cities is often more extensive than that of households in urban counties and cities. Therefore, the average living area per person (LAP) has a spatial spillover effect on household consumption spending.

Food plays an important part in household consumption, and high consumption on food in a household crowds out household consumption in other fields. Mottaleb et al. (2017) have stated that food away from home (FAFH) consumption is an established phenomenon in households in developed countries, but in many middle-income and rapidly developing countries, FAFH is a growing phenomenon. It can be seen from previous studies that the characteristics of different households exhibit different economic strengths, such as the difference in the number of employed persons in each household, the difference in disposable income of each household, the difference in self-owned houses (whether they own or not and the size of the living area), and the difference in the cost of FAFH, which further affect the consumption spending of each household. Therefore, this study established the following hypotheses, of which H2a is the direct effect, and H2b is the spatial spillover effect.

**Hypothesis 2a.** *Household characteristics in local counties and cities (average disposable income per household, average number of employed persons per household, percentage of self-owned housing, average living area per capita, and percentage of food spending in total consumption spending) have a significant influence on average consumption spending per household in local counties and cities.*

**Hypothesis 2b.** *Household characteristics in neighboring counties and cities (average disposable income per household, average number of employed persons per household, percentage of self-owned housing, average living area per capita, and percentage of food spending in total consumption spending) have a significant influence on the average consumption spending per household in local counties and cities.*

### 3. Methodology

#### 3.1. Data and Sample

This study used panel data from 22 counties and cities in Taiwan from 2001 to 2020 as samples. The data were mainly obtained from The Query System for Important Statistical Indicators of Counties and Cities of National Statistics, R.O.C. (Taiwan). Table 1 shows the statistical data of Taiwan's average consumption spending (NT$) per household from 2000 to 2020. In 2000, Taiwan's average consumption spending per household was NT$603,772. By 2020, Taiwan's average consumption spending per household was NT$737,768, where US$1 was NT$29.5.

**Table 1.** Taiwan's average consumption spending per household (NT$) from 2001 to 2020.

| Year | Average Consumption Spending per Household | Year | Average Consumption Spending per Household | Year | Average Consumption Spending per Household |
|------|--------------------------------------------|------|--------------------------------------------|------|--------------------------------------------|
| 2000 | 603,772 | 2007 | 649,905 | 2014 | 682,220 |
| 2001 | 595,870 | 2008 | 635,019 | 2015 | 684,841 |
| 2002 | 606,897 | 2009 | 646,966 | 2016 | 706,662 |
| 2003 | 606,895 | 2010 | 646,182 | 2017 | 723,078 |
| 2004 | 631,186 | 2011 | 661,160 | 2018 | 723,957 |
| 2005 | 637,384 | 2012 | 662,392 | 2019 | 740,302 |
| 2006 | 650,354 | 2013 | 683,004 | 2020 | 737,768 |

*3.2. Research Variables*

The definitions and explanations of dependent variables and independent variables in the empirical model of this study are as follows.

### 3.2.1. Dependent Variable

Average consumption spending per household (NTD) ($CS_{it}$): The average consumption spending per household in the $t$th year of the $i$th county or city in Taiwan. Average consumption spending per household is defined as the total consumption spending per household. The calculation formula: (total consumption spending ÷ total number of households).

### 3.2.2. Independent Variables Related to Fiscal Expenditure

1. Economic development expenditure (million NTD) ($EDE_{it}$): The economic development expenditure in the $t$th year of the $i$th county or city in Taiwan. Economic development expenditure refers to the expenditure on agriculture, industry, transportation, and other economic services of the county and city.

2. Expenditure on education, science, & culture (million NTD) ($EESC_{it}$): The expenditure on education, science, & culture in the $t$th year of the $i$th county or city in Taiwan. Expenditure on education, science, & culture refers to the expenditure and subsidies on education, science, culture, and other undertakings of the county and city.

### 3.2.3. Independent Variables Related to Household Characteristics

1. Average disposable income per household (NTD) ($DI_{it}$): The average disposable income per household in the $t$th year of the $i$th county or city in Taiwan. The calculation formula is: (disposable income ÷ total households).

2. Average number of employed persons per household (persons) ($EPH_{it}$): The average number of employed persons per household in the $t$th year of the $i$th county or city in Taiwan. The calculation formula is: (total employees ÷ total households).

3. Home ownership percentage (%) ($HOP_{it}$): The home ownership percentage in the $t$th year of the $i$th county or city in Taiwan. The definition of home ownership, in 2009 and before, refers to a situation in which "the ownership of the existing house belongs to any of the household members or their immediate relatives". Since 2010, in line with the definition of population and housing census, this has been amended to "owned by the household member who is a regular resident". The calculation formula is: (number of self-owned houses ÷ total number of houses) × 100.

4. Average living area per person ($LAP_{it}$): The average ping of housing per person in the $t$th year of the $i$th county or city in Taiwan. The calculation formula is: (average ping per household ÷ average number of the household member).

5. Percentage of food spending (excluding FAFH) to consumption spending (%) ($FSCS_{it}$): refers to the percentage of food spending (excluding FAFH) to consumption spending in the $t$th year of the $i$th county or city in Taiwan. Food away from home (FAFH) refers to expenses for weddings, birthday celebrations, funerals, sacrificial feasts (restaurants and catering), and meals away from home.

### 3.3. Empirical Model

LeSage and Pace (2009) proposed the spatial Durbin model (SDM), which includes the lagged variables of dependent variables as well as that of independent variables. The spatial Durbin model constructed in this study is as follows.

$$
CS_{it} = \rho \sum_{j=1}^{N} W_{ij} CS_{jt} + \alpha + \beta_1 EDE_{it} + \beta_2 EESC_{it} + \beta_3 DI_{it} + \beta_4 EPH_{it} + \beta_5 HOP_{it} + \beta_6 LAP_{it} + \beta_7 FSCS_{it}
$$
$$
+ \theta_1 \sum_{j=1}^{N} W_{ij} EDE_{jt} + \theta_2 \sum_{j=1}^{N} W_{ij} EESC_{jt} + \theta_3 \sum_{j=1}^{N} W_{ij} DI_{jt} + \theta_4 \sum_{j=1}^{N} W_{ij} EPH_{jt} + \theta_5 \sum_{j=1}^{N} W_{ij} HOP_{jt}
$$
$$
+ \theta_6 \sum_{j=1}^{N} W_{ij} LAP_{jt} + \theta_7 \sum_{j=1}^{N} W_{ij} FSCS_{jt} + \mu_i + \varepsilon_{it} \, i \neq j
$$

where, $CS_{it}$ is a dependent variable, $i$ and $j$ are counties and cities in Taiwan, and $t$ is the year (t = 2001–2020). $W_{ij}$ is the spatial weight matrix, which is a square matrix symmetrical in the upper right and lower left, and the number of columns and rows is equal to the number of counties and cities (22 counties and cities in total in this study). The contiguity matrix was used to define the proximity relationship. If two counties and cities are defined by "proximity", the value is 1 and otherwise 0. The diagonal is also 0 (one county or city cannot be adjacent to itself).

The variable $\rho$ is the spatially lagged coefficient and is the spatial autocorrelation coefficient of the dependent variable, reflecting the direction and degree of influence of the dependent variable $CS_{jt}$ of the neighboring county or city on the local county or city $CS_{it}$. By verifying the spatially lagged coefficient $\rho$, the spillover effect of neighboring counties and cities on local counties and cities can be further explored. The variable $\rho$ can significantly demonstrate an obvious spatial dependence between dependent variables, where $\rho \neq 0$ means a spatial relationship with neighboring counties and cities, and $\rho > 0$ indicates a positive space spillover effect and that a positive effect of space spillover has been created. The value of $\rho$ reflects the interaction degree of spatial diffusion or spatial spillover between counties and cities.

$W_{ij} CS_{jt}$ is the spatial autocorrelation term matrix of dependent variables and is the endogenous variable, showing the influence of $CS_{jt}$ of $j$ county and city adjacent to $CS_{it}$ of $i$ county and city. $\alpha$ is a constant term, and $\beta$ is the coefficient to be estimated and represented the original effect coefficient.

$EDE_{it}$, $EESC_{it}$, $DI_{it}$, $EPH_{it}$, $HOP_{it}$, $LAP_{it}$, and $FSCS_{it}$ are independent variables. $\beta_1 EDE_{it}$, $\beta_2 EESC_{it}$, $\beta_3 DI_{it}$, $\beta_4 EPH_{it}$, $\beta_5 HOP_{it}$, $\beta_6 LAP_{it}$, and $\beta_7 FSCS_{it}$ reflect the influence of all independent variables on $CS_{it}$. The coefficient $\theta$ is to be estimated and is the spatial autocorrelation coefficient of the independent variable, reflecting the influence of all independent variables in the neighboring counties and cities on the $CS_{it}$ of the local county and city—that is, representing the influence of all independent variables $EDE_{jt}$, $EESC_{jt}$, $DI_{jt}$, $EPH_{jt}$, $HOP_{jt}$, $LAP_{it}$, and $FSCS_{it}$ of the counties and cities that have a spatial association with the local county or city on the dependent variable $CS_{it}$ of the local county and city. Positive $\theta$ indicates that the neighboring county effect has a positive effect on the dependent variable $CS_{it}$, and that there is a spillover effect between neighboring counties and cities. Negative $\theta$ indicates a competitive effect between neighboring counties and cities.

$W_{ij} EDE_{jt}$, $W_{ij} EESC_{jt}$, $W_{ij} DI_{jt}$, $W_{ij} EPH_{jt}$, $W_{ij} HOP_{jt}$, $W_{ij} LAP_{jt}$, and $W_{ij} FSCS_{jt}$ are the spatial autocorrelation term matrices of the independent variables. They represent the spatial effect of all independent variables and show the influence of all independent variables of the neighboring counties and cities on the $CS_{it}$ of local counties and cities. $\mu_i$ is the spatial (individual) effect and the individual effect of $i$ county or city. Lastly, $\varepsilon_{it}$ is the independent and identically distributed random error term and the spatially autocorrelated error term.

### 3.4. Spatial Weight Matrix

This study defined a binary symmetric spatial weight matrix $W_{n \times n}$ to express the spatial proximity of n counties and cities.

$$W_{ij} = \begin{bmatrix} w_{11} & w_{12} & \cdots & w_{1n} \\ w_{21} & w_{22} & \cdots & w_{2n} \\ \vdots & \vdots & \cdots & \vdots \\ w_{n1} & w_{n2} & \cdots & w_{nn} \end{bmatrix}$$

The identification methods of contiguity of spatial units could be divided into rook contiguity, bishop contiguity, and queen contiguity. Rook contiguity means that there is contact between two spatial boundaries. Bishop contiguity refers to diagonal adjacency. Queen contiguity refers to the counties and cities that have contact in both boundaries and diagonal areas (Sawada 2004). Queen contiguity was used to define spatial contiguity in this study.

This study used adjacency rules to establish a spatial weight matrix, which is defined as:

$$W_{ij} = \begin{cases} 1, & \textit{regions i and j are neighboring regions} \\ 0, & \textit{regions i and j are not neighboring regions} \end{cases}$$

Taiwan has three island counties, which have no neighboring relationship with other counties, and so their value is 0 (Huang et al. 2022).

## 4. Results

### 4.1. Descriptive Statistics

Table 2 shows the variables in this research model. Table 3 lists the correlation analysis results. The variables used in the table are raw data.

**Table 2.** Summary of descriptive statistics.

| Variables | Obs. | Mean | Std. Dev. | Min. | 25th Percentile | Median | 75th Percentile | Max. |
|---|---|---|---|---|---|---|---|---|
| CS | 462 | 662,657.90 | 146,314.08 | 390,512.00 | 558,868 | 635,692 | 738,518 | 1,152,501.00 |
| EDE | 462 | 6894.38 | 7291.91 | 541.91 | 2446.65 | 4127.63 | 7720.55 | 46,357.08 |
| ESCE | 462 | 14,560.91 | 15,980.28 | 400.57 | 4537.28 | 7299.64 | 18,247.86 | 73,161.88 |
| DI | 462 | 875,352.29 | 174,144.36 | 568,409.00 | 746,442 | 836,491 | 967,197 | 1,422,856.00 |
| EPH | 462 | 1.42 | 0.18 | 0.94 | 1.30 | 1.44 | 1.55 | 1.88 |
| HOP | 462 | 86.79 | 4.42 | 70.03 | 83.80 | 86.58 | 89.98 | 95.94 |
| LAP | 462 | 14.58 | 2.82 | 8.55 | 12.58 | 14.78 | 16.41 | 23.12 |
| FSCS | 462 | 15.92 | 2.06 | 10.83 | 14.63 | 15.64 | 17.24 | 23.79 |

Note: CS: Average consumption spending per household (NTD); EDE: Economic development expenditure (million NTD); EESC: Expenditure on education, science, & culture (million NTD); DI: Average disposable income per household (NTD); EPH: Average number of employed persons per household (persons); HOP: Home ownership percentage (%); LAP: Average living area per person (*ping*): FSCS: Percentage of food spending (excluding FAFH) to consumption spending (%).

**Table 3.** Pearson correlation analysis.

| | CS | EDE | EESC | DI | EPH | HOP | LAP | FSCS |
|---|---|---|---|---|---|---|---|---|
| CS | 1 | | | | | | | |
| EDE | 0.459 ** | 1 | | | | | | |
| EESC | 0.557 ** | 0.897 ** | 1 | | | | | |
| DI | 0.911 ** | 0.405 ** | 0.486 ** | 1 | | | | |
| EPH | 0.302 ** | 0.132 ** | 0.198 ** | 0.283 ** | 1 | | | |
| HOP | −0.258 ** | −0.282 ** | −0.287 ** | −0.319 ** | 0.290 ** | 1 | | |
| LAP | −0.231 ** | −0.269 ** | −0.371 ** | −0.180 ** | −0.295 ** | 0.153 ** | 1 | |
| FSCS | −0.399 ** | −0.149 ** | −0.201 ** | −0.400 ** | −0.133 ** | 0.034 | 0.100 * | 1 |

Note: * $p < 0.05$, ** $p < 0.01$.

### 4.2. Spatial Autocorrelation Test Results

Among various spatial autocorrelation indicators, the statistical power of global spatial autocorrelation (Moran's I) is the best (Walter 1992). Therefore, Moran's I is the most widely used spatial autocorrelation indicator (Cliff and Ord 1973, 1981). The range of Moran's I is [−1, 1], and the closer its value approaches 1, the stronger is the degree of positive spatial autocorrelation. The closer its value approaches -1, the stronger the degree of negative spatial autocorrelation is (Moran 1950). Moran's I analysis aims at an overall space-related situation, understanding whether data have agglomeration phenomenon in spatial characteristics, but it cannot analyze regional changes and point out the spatial distribution of hot spots.

As seen from Table 4, the values of Moran's I all reach a significant level of 0.05, indicating that the average consumption spending per household of each county and city from 2000 to 2020 has a significantly positive spatial autocorrelation, and the average consumption spending per household shows a strong spatial agglomeration feature. They also show that the average consumption spending per household in Taiwan presents a geographical agglomeration phenomenon.

**Table 4.** Spatial autocorrelation indicators from 2000 to 2020.

| Year | Moran's I | | Year | Moran's I | | Year | Moran's I | |
| --- | --- | --- | --- | --- | --- | --- | --- | --- |
| | I | *p*-Value | | I | *p*-Value | | I | *p*-Value |
| 2000 | 0.329 | 0.021 | 2007 | 0.417 | 0.006 | 2014 | 0.499 | 0.001 |
| 2001 | 0.357 | 0.015 | 2008 | 0.557 | 0.000 | 2015 | 0.452 | 0.004 |
| 2002 | 0.386 | 0.009 | 2009 | 0.595 | 0.000 | 2016 | 0.459 | 0.003 |
| 2003 | 0.405 | 0.007 | 2010 | 0.542 | 0.001 | 2017 | 0.467 | 0.003 |
| 2004 | 0.475 | 0.002 | 2011 | 0.550 | 0.000 | 2018 | 0.449 | 0.004 |
| 2005 | 0.579 | 0.000 | 2012 | 0.483 | 0.002 | 2019 | 0.384 | 0.012 |
| 2006 | 0.417 | 0.005 | 2013 | 0.429 | 0.005 | 2020 | 0.391 | 0.010 |

Anselin (1995) divided local indicators of spatial association (LISA) into four quadrants according to the degree of spatial agglomeration, so as to indicate the spatial relationship between the local county and city and the neighboring counties and cities. LISA provides an explanatory model of spatial interaction. Moran's I cannot show the spatial agglomeration between counties and cities, nor the spatial autocorrelation characteristics of counties and cities, but LISA can make up for the deficiency. The statistical verification results of LISA are presented in the form of maps. Different color blocks distinguish the spatial autocorrelation of counties and cities that have reached a significant level according to their categories, so as to observe the changes in the LISA agglomeration phenomenon at different time points and understand the changes in spatial structure with time.

In LISA's clustering map, the first quadrant and the third quadrant are the zones where similar values agglomerated. The first quadrant represents high-value agglomeration, while the third quadrant represents low-value agglomeration. In the first quadrant, there is a hot zone where the average consumption spending per household in the local and neighboring counties and cities were all high, and it is expressed as High-High (HH). In the third quadrant, there is a cold zone where the average consumption spending per household in the local and neighboring counties and cities were all low, and it is represented as Low-Low (LL). The second quadrant and the fourth quadrant are the zones where different values agglomerated and the negative areas spatially autocorrelated. In the second quadrant, low values are surrounded by high values, and the average consumption spending per household in a local county or city was low, while that in the surrounding counties or cities was high. This is represented as Low-High (LH). In the fourth quadrant, high values are surrounded by low values, and the average consumption spending per household in a local county or city was high, while that in the surrounding counties or cities was low. This is represented as High-Low (HL). A significantly negative spatial autocorrelation means

that the average consumption spending per household in the local county or city was very different from that in the neighboring counties or cities, which is called a spatial outlier.

Figure 1 shows the LISA cluster diagrams of average consumption spending per household in Taiwan from 2000 to 2020. This study illustrated the change in average consumption spending per household in Taiwan by using LISA cluster diagrams of six stages. These six phases represented years in which presidential elections were held. From 2000 to 2020, Taiwan experienced two political party changes. Some counties and cities faced changes due to the rotation of the political power. For example, Kaohsiung City is a traditional voting base of the Democratic Progressive Party (DPP), and this city was a cold zone (denoted by LL) when DPP was in power, but was transformed into an HL zone in 2012 after the change of political parties in 2008. Some counties and cities, such as New Taipei, Miaoli, Hsinchu, and Nantou, remained unchanged no matter which party was in power in the central government.

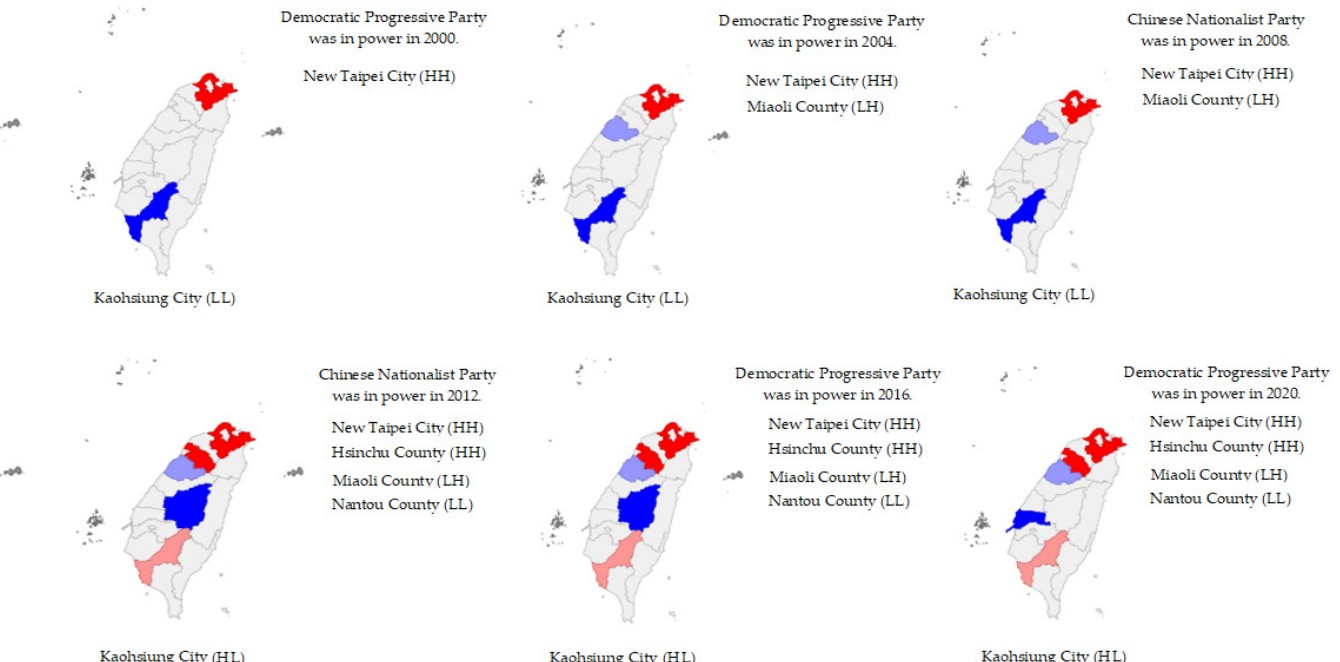

**Figure 1.** LISA cluster diagrams of changes in average household consumption spending in Taiwan from 2000 to 2020.

### 4.3. Hausman Test Results

LeSage and Pace (2009) and Elhorst (2010) pointed out that the spatial Durbin model could be simplified as the spatial lag model or the spatial error model. The model was selected through the verification of the following hypothesis. $H_0$: $\theta = 0$; if the original hypothesis was rejected, the spatial Durbin model could not be simplified to the spatial lag model. Thus, it was more appropriate to choose the spatial Durbin model. $H_0$: $\theta + \rho\beta = 0$; if the original hypothesis was rejected, the spatial Durbin model could not be simplified as the spatial error model. Thus, it was more appropriate to select the spatial Durbin model. Between the spatial lag model and the spatial Durbin model, the Wald test indicated $x^2 = 34.36$, $p < 0.001$, and the likelihood-ratio test indicated $x^2 = 46.07$, $p < 0.001$. Therefore, it was more appropriate to select the spatial Durbin model. Between the spatial error model and the spatial Durbin model, the Wald test indicated $x^2 = 47.17$, $p < 0.001$, and the likelihood-ratio test indicated $x^2 = 47.83$, $p < 0.001$. Therefore, it was more appropriate to select the spatial Durbin model.

This study used the Hausman test proposed by Hausman (1978) to determine whether the random-effects model or fixed-effects model is more suitable for analysis. Table 5 shows the results of the Hausman test. The Hausman test of the spatial Durbin model for

space fixed effects and the spatial Durbin model for random effects calculates $x^2 = 36.89$, $p < 0.001$, thus rejecting the null hypothesis (random-effects model). The Hausman test of the spatial Durbin model for time fixed effects and the spatial Durbin model for random effects calculates $x^2 = 127.95$, $p < 0.001$, also rejecting the null hypothesis (random-effects model). The Hausman test of the spatial Durbin model for space and time fixed effects and the spatial Durbin model for random effects calculates $x^2 = 16.67$, $p < 0.01$, also rejecting the null hypothesis (random-effects model). Therefore, the fixed-effects model, the time fixed-effects model, and the space and time fixed-effects model are all more suitable than the random-effects model.

**Table 5.** Hausman test results.

| | Hausman Test | |
|---|---|---|
| | $x^2$ | *p*-**Value** |
| SDM with spatial fixed-effects vs. SDM with random-effects | 36.89 | 0.0000 |
| SDM with time fixed-effects vs. SDM with random-effects | 127.95 | 0.0000 |
| SDM with spatial and time fixed-effects vs. SDM with random-effects | 16.67 | 0.0022 |

*4.4. Spatial Durbin Model Analysis Results*

Table 6 shows the analysis results of four spatial Durbin models. As for the overall goodness-of-fit, the spatial econometric model uses maximum likelihood estimation (MLE), and so the model can obtain the maximum similar and non-linear test value and maximize its coefficient. Only goodness-of-fit tests based on non-linear principles, such as log likelihood (LIK), Akaike information criterion (AIC) of Akaike (1973), or Schwartz Bayesian information criterion (SBC or BIC) of Schwarz (1978) can be used as indicators of fitness verification. Versus the maximum log likelihood value or minimum AIC value, the BIC value is the best model. If the value of AIC and BIC is used to evaluate the goodness-of-fit of the model, the smaller the value is, the higher is the fitness of the model.

According to the results of the Hausman test, this study must choose between Model 1, Model 2, and Model 3. The log-likelihood, AIC, and BIC values of Model 3 were all better than those of Model 1 and Model 2. However, the spatially lagged coefficient $\rho$ of Model 3 did not reach a significant level ($\rho = 0.06$, $p = 0.289$). Therefore, this study finally decided to adopt Model 1 (spatial Durbin model of spatial fixed effects) as the basis of analysis. The log likelihood, AIC, and BIC values of Model 1 were all better than those of Model 2. The spatially lagged coefficient $\rho$ of Model 1 reached a significant level ($\rho = 0.20$, $p < 0.001$), indicating that there is significant spatial autocorrelation between the distributions of average consumption spending per household in counties and cities, which again confirms the rationality of incorporating spatial effects into the econometric model.

**Table 6.** Spatial Durbin model analysis results.

| Variables | Model 1 SDM with Spatial Fixed Effects | | Model 2 SDM with Time Fixed Effects | | Model 3 SDM with Spatial and Time Fixed Effects | | Model 4 SDM with Random Effects | |
|---|---|---|---|---|---|---|---|---|
| | Coefficient | *p* | Coefficient | *p* | Coefficient | *p* | Coefficient | *p* |
| EDE | −0.31 | 0.470 | −0.58 | 0.298 | −0.63 | 0.132 | −0.35 | 0.428 |
| EESC | 0.15 | 0.736 | 0.77 * | 0.011 | 0.40 | 0.351 | 0.46 | 0.262 |
| DI | 0.54 *** | 0.000 | 0.78 *** | 0.000 | 0.54 *** | 0.000 | 0.59 *** | 0.000 |
| EPH | 97,673.82 *** | 0.000 | −62,410.52 *** | 0.000 | 100,764.50 *** | 0.000 | 71,187.76 ** | 0.002 |
| HOP | −139.18 | 0.780 | 1121.95 | 0.034 | −579.36 | 0.256 | −287.73 | 0.570 |
| LAP | 3839.11 ** | 0.003 | −3968.27 ** | 0.001 | 1682.71 | 0.304 | 2306.16 | 0.089 |
| FSCS | −1047.41 | 0.326 | 3351.63 ** | 0.001 | −741.96 | 0.478 | −752.29 | 0.486 |
| W × EDE | 0.92 | 0.139 | −0.45 | 0.589 | −0.12 | 0.848 | 0.74 | 0.244 |
| W × EESC | 1.06 | 0.098 | 0.03 | 0.943 | 2.09 ** | 0.002 | 0.69 | 0.258 |
| W × DI | −0.20 *** | 0.000 | −0.14 *** | 0.000 | −0.09 | 0.135 | −0.18 ** | 0.001 |
| W × EPH | −35,056.15 | 0.310 | 13,246.75 | 0.694 | −12,424.23 | 0.719 | −30,375.68 | 0.352 |
| W × HOP | 1230.91 | 0.164 | 1919.00 * | 0.016 | 148.49 | 0.891 | 1007.20 | 0.115 |
| W × LAP | 8807.69 *** | 0.000 | 7678.66 *** | 0.000 | 7741.62 *** | 0.001 | 5621.29 ** | 0.005 |
| W × FSCS | 1292.51 | 0.541 | −4003.87 ** | 0.008 | 4785.39 * | 0.025 | −2135.21 | 0.252 |
| Constant | | | | | | | −48,320.74 | 0.429 |
| n | | 462 | | 462 | | 462 | | 462 |
| Spatial $\rho$ | 0.20 *** | 0.000 | 0.21 *** | 0.000 | 0.06 | 0.289 | 0.23 *** | 0.000 |
| within $R^2$ | | 0.7835 | | 0.7456 | | 0.7609 | | 0.7793 |
| between $R^2$ | | 0.6524 | | 0.8197 | | 0.7698 | | 0.9046 |
| overall $R^2$ | | 0.6731 | | 0.8061 | | 0.7683 | | 0.8787 |
| Log-likelihood | | −5369.2162 | | −5502.5384 | | −5342.3046 | | −5427.4673 |
| AIC | | 10,756.43 | | 11,023.08 | | 10,702.61 | | 10,874.93 |
| BIC | | 10,793.65 | | 11,060.30 | | 10,739.83 | | 10,916.29 |

Note: * $p < 0.05$; ** $p < 0.01$; *** $p < 0.001$.

### 4.5. Decomposition Results of the Spatial Durbin Model with Spatial Fixed Effects

LeSage and Pace (2009) have pointed out that in the spatial Durbin model where spatial interaction effects are taken into consideration, if regression results of spatial estimation parameters are directly used to determine whether there is a spatial spillover effect, then the existence of feedback effects (FE) may lead to incorrect conclusions. Due to the inclusion of spatially lagged independent variables and dependent variables in the spatial Durbin model, the estimated results cannot directly reflect their marginal effects, and it is difficult to accurately measure the direct impact of independent variables on dependent variables (Elhorst 2010). Therefore, the spatial Durbin model must be divided into direct effects, indirect effects, and total effects. Table 7 shows the decomposition results of the spatial Durbin model with spatial fixed effects. The factors related to family mutual characteristics are illustrated as follows.

**Table 7.** Direct, indirect, and total effects of SDM with spatial fixed effects.

| Variables | Direct Effect | | Indirect Effect | | Total Effect | |
|---|---|---|---|---|---|---|
| | Coefficient | *p*-Value | Coefficient | *p*-Value | Coefficient | *p*-Value |
| EDE | −0.25 | 0.570 | 0.87 | 0.164 | 0.61 | 0.488 |
| EESC | 0.18 | 0.678 | 1.19 | 0.066 | 1.37 | 0.076 |
| DI | 0.54 *** | 0.000 | −0.09 *** | 0.036 | 0.45 *** | 0.000 |
| EPH | 95,654.02 *** | 0.000 | −15,854.58 | 0.647 | 79,799.44 * | 0.036 |
| HOP | −71.47 | 0.882 | 1321.16 | 0.174 | 1249.69 | 0.249 |
| LAP | 4282.79 *** | 0.000 | 9811.18 *** | 0.000 | 14,093.98 *** | 0.000 |
| FSCS | −990.72 | 0.382 | 1212.04 | 0.576 | 221.32 | 0.928 |

Note: * $p < 0.05$; *** $p < 0.001$.

The spatial effects of average disposable income per household (NT$) (DI). Direct effect: An increase in disposable income per household leads to an increase in consumption spending per household in local counties and cities. For every unit increase in the average disposable income per household in a local county or city, the average consumption spending per household in a local county or city will increase by 0.54 units. Indirect effects: The increase in average disposable income per household has negative spillover effects on average consumption spending per household in neighboring counties and cities. For every unit increase in the average disposable income per household in the neighboring counties and cities, the average consumption spending per household in the local county will decrease by 0.09 units.

The spatial effects of the average number of employed persons per household (person) (EPH). Direct effect: An increase in the average number of persons employed per household will lead to an increase in average consumption spending per household in local counties and cities. For every unit increase in the average number of employed persons per household in a local county or city, the average consumption spending per household in a local county or city will increase by 95,654 units. Indirect effect: An increase in average household employment does not have a significant effect on average household spending in neighboring counties and cities.

Spatial effects of average living area per person (ping) (LAP). Direct effect: An increase in living area per person will lead to an increase in consumption spending per household in local counties and cities. For every unit increase in the average living area per person in a local county or city, the average consumption spending per household in a local county or city will increase by 4282 units. Indirect effect: The growth of the average living area per person has a positive spillover effect on the average consumption spending per household in neighboring counties and cities. For every unit increase in the average living area per person in neighboring counties and cities, the average consumption spending per household in local counties and cities will increase by 9811 units.

## 5. Conclusions and Policy Implications

### 5.1. Conclusions

From the perspective of spatial interdependence, Moran's I showed that the average consumption spending per household in Taiwan does show spatial agglomeration on the whole, but Moran's I analysis cannot be used to further analyze the regional spatial correlation patterns of different geographical locations. This study used local indicators of spatial association (LISA) to analyze the agglomeration of specific counties and cities, then examined the central ruling parties in the agglomeration years, and found that the different central ruling parties also have an impact on the agglomeration of specific counties and cities. Most of the counties and cities in the hot spots with high value in agglomeration are located in the northern part of Taiwan, while most of the counties and cities in the cold spots with a low value in agglomeration are located in central and southern Taiwan. The results show that there was an uneven difference in consumption spending between the northern and the southern regions of Taiwan. In terms of the transition paths of counties and cities, Kaohsiung City was originally located in the third quadrant (LL) of the LISA clustering diagram, and during that period DPP was in charge of the central government. Over time, Kaohsiung City located to the fourth quadrant (HL), when the KMT was in charge of the central government. The transition path of Kaohsiung City belongs to the transition type of the county or city itself (LL→HL). Some counties and cities remain unchanged regardless of which party was in power. The transition path belongs to the path where the county or city is at the same level and all counties and cities remain unchanged, such as New Taipei City, Miaoli County, Hsinchu County, and Nantou County.

In terms of direct effects, the average consumption spending per household in a local county or city is influenced by household characteristics, including average disposable income per household, average number of employees per household, and average living area per capita. In terms of the impact of average living area per capita, the exorbitant housing

price in Taiwan means urban residents have to pay a large quantity of money in order to buy a spacious and comfortable house. Therefore, the proportion of housing costs in consumption is increasingly high. In terms of spatial spillover effects, the average consumption spending per household in a local county or city is influenced by household characteristics in the neighboring counties and cities, including average disposable income per household and average living area per capita. The results show that household characteristics have direct effects and spatial spillover effects on the average consumption spending per household, which is similar to the results of previous studies. The consumption spending of each household is mainly influenced by household characteristics.

The local economic development expenditure and local expenditure on education, science, and culture adopted in this study had surprisingly no significant impact on the average consumption spending per household in counties and cities. In terms of the statistical significance of spatial spillover effects and total effects of local expenditure on education, science and culture, although the value did not reach the significant level of 0.05, it did reach the level of $p < 0.1$. In this study, the spatial effects were incorporated into the spatial econometric model for analysis, which was somewhat different from the analysis of the traditional econometric model in the past. It is suggested that subsequent researchers collect a longer study period for verification or adopt other fiscal expenditure variables, which may lead to different research conclusions.

### 5.2. Policy Implications

As Taiwan's economy has developed by leaps and bounds, the life of its residents is changing constantly. The result of economic development is embodied in the consumption spending of the residents. Consumption spending is a big contributor to economic growth. In order to have a deeper understanding of the current economic status of Taiwan, it is essential to study the influencing factors of residents' consumption spending in the counties and cities. Although Taiwan's economy is steadily developing, differences among counties and cities still exist and are difficult to overcome. The differences also cause an economic structure imbalance in Taiwan, which slows down economic development and hinders economic growth. Nowadays, narrowing the differences between counties and cities and narrowing the wealth gap between urban and rural areas have become two of the major issues that Taiwan should tackle. For instance, the education expenditure in Taiwan is a big burden for residents, including fees for after-school tuition and learning various talents. From the perspective of fiscal expenditure, if the government could increase the expenditure on education, science, and culture or reduce the burden of residents by subsidizing expenditure on education, which could increase consumption in the domestic market by allowing people to spend elsewhere, then economic growth would be achieved. The empirical results of this study help us to understand the spatial dependence of consumption spending among counties and cities in Taiwan. The results herein could be used as a reference for the government to address the gap between urban and rural areas.

### 5.3. Research Limitations

This section presents some research limitations in this study. First, the spatial effect includes spatial dependence (autocorrelation) and spatial heterogeneity. Spatial heterogeneity mainly consists of the heterogeneity of spatial units in shape and size and the non-stationary structure of economic phenomena in space. In this study, spatial Durbin model analysis could only solve spatial autocorrelation but could not wholly solve spatial heterogeneity. Second, in terms of research samples, there are only 22 administrative regions in Taiwan. This number is relatively small compared to other countries or regions, which may lead to fewer research samples. Third, in terms of the research period, this study examined data from 2000 to 2020. It is difficult to establish and collect panel data. If some variables are not gathered in the government database, or the data in some counties and cities are missing, it will lead to difficulties in data collection and analysis. Furthermore, the timing of information release from the Taiwan government also caused some limitations. The

government has not released some variables in the statistics of counties and cities from 2021. As this study could not access the data in 2021, it took 2000 to 2020 as the research period.

### 5.4. Future Research Directions

This study suggests several future research directions. First, due to the spatial correlation or spatial heterogeneity among counties and cities, the impact of fiscal expenditures and household characteristics on each county and city is different. This study adopted the spatial Durbin model to empirically verify the effect of fiscal expenses and household characteristics on average household consumption spending. However, we suggest other variables are worth discussing, such as local fiscal expenditure variables. In terms of the division of expenditure structure, the expenditure structure in Taiwan's local governments can be divided into productive expenditures (educational, scientific, and cultural expenditures, and economic development expenditures), non-productive expenditures (general government affairs expenditures, social welfare expenditures, and retirement pension expenditures) and other expenditures (community development and environmental protection expenditures, police administration expenditures, debt expenditures, and assistance and auxiliary expenditures). It is suggested that subsequent researchers may find suitable variables from the literature or other theories for modeling and analysis. Second, it is recommended that subsequent research extends this topic to include spatial heterogeneity for the analysis with other, more complex, spatial econometric models. To solve the issue of spatial heterogeneity, a geographically weighted regression model (GWR) can be adopted for processing (Brunsdon et al. 1996).

**Author Contributions:** Conceptualization, H.-C.H. and C.-L.Y.; methodology, H.-C.H.; software, T.-H.L.; validation, T.-H.L. and C.-L.Y.; formal analysis, C.-L.Y. and T.-H.L.; investigation, H.-C.H. and C.-L.Y.; resources, T.-H.L.; data curation, H.-C.H. and T.-H.L.; writing—original draft preparation, C.-L.Y. and T.-H.L.; writing—review and editing, H.-C.H.; supervision, H.-C.H.; All authors have read and agreed to the published version of the manuscript.

**Funding:** This research received no external funding.

**Informed Consent Statement:** Not applicable.

**Data Availability Statement:** Not applicable.

**Conflicts of Interest:** The authors declare no conflict of interest.

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
