# Peer review of "The Spatial Spillover Effects of Fiscal Expenditures and Household Characteristics on Household Consumption Spending: Evidence from Taiwan"

_economies, doi:10.3390/economies10090227_

Round 1

Reviewer 1 Report

I believe the paper is acceptable for publication after some minor revisions, please find comments in the attachment.

Author Response

Reviewer 1

One problem I had was that the paper did not provide a stronger economic motivation for the spatial effects of the household characteristics (see Hypothesis 2b). A better motivated analysis in this regard would also help to understand the indirect effects of DI and LAP.

Response:

Your opinions are greatly valued. If this paper can be successfully accepted and published in Economies, we would like to extend our warmest gratitude to you. In the amended paper, a clear description of the relationship between average living area per person (LAP) and disposable income (DI), as well as average consumption spending (CS), have been made for H2. The amendments are as follows:

The size of the average living area per person (LAP) is related to housing prices. For example, Suari-Andreu (2021) and Carroll, Otsuka, and Slacalek (2011) found a strong correlation between housing price evolution and household consumption. Since housing prices in Taiwan have always been high, the cost of owning a spacious house is bound to be high. Excessively high housing prices will undoubtedly affect household consumption spending. Compared to other rural counties and cities, the average living area per person (LAP) of households in several large urban counties and cities in Taiwan, such as Taipei City, New Taipei City, or Taichung City, is much smaller than that in rural counties and cities, such as Changhua County, Chiayi County, and Pingtung County, with the same housing price. Household spending on purchases or rentals accounts for a high proportion of consumer spending. The desire for a larger average living area per person (LAP) may affect household spending. Typically, the neighborhoods surrounding urban counties and cities are rural. High household consumption spending in urban counties and cities causes many people to move to neighboring areas with relatively lower expenditures. The average living area per person (LAP) of households in these rural counties and cities is often more extensive than that of households in urban counties and cities. Therefore, the average living area per person (LAP) has a spatial spillover effect on household consumption spending.

Reviewer 2 Report

A spatial econometric model, the spatial Durbin model, was used to analyze the spatial spillover effects of fiscal expenditures and household characteristics on household consumption spending using Taiwan county-level and municipal panel data from 2000 to 2020. The empirical results show that the average consumption spending per household of all counties and cities in Taiwan presents spatial auto-correlation and the agglomeration, and the average consumption spending per household is influenced by household characteristics of the neighboring counties and cities. This is a well-written academic paper, but there are still some problems. The details are as follows:

1. The structure of the chapter introduction” is not clear, so the authors should analyze the main purpose and research background of this study, the influencing factors of household consumption spending and the structure of the article from levels and paragraphs.

2. In the chapter 2.1. Application of the Spatial Econometric Model in Consumption Spending, the author mentioned: there are relatively few studies on the spatial measurement of average household consumption spending. What are the main research methods used in the existing literature? Besides the research method, what are the main marginal contributions of this paper.

3. Queen contiguity was used to define spatial weights matrix. It is suggested that the authors consider the distance and economic gap between counties and cities when constructing the spatial weight matrix.

4. What is the author's criterion for using a spatial Durbin model rather than a spatial autocorrelation or spatial lag model? Have relevant tests been carried out.

5. Except for conclusions and policy implications, there is more room to focus on the limitations and research outlook. 

Author Response

Reviewer 2

We appreciate your opinions. If this paper can be successfully accepted and published in Economies, we would like to extend our warmest gratitude to you.

  1. The structure of the chapter “introduction” is not clear, so the authors should analyze the main purpose and research background of this study, the influencing factors of household consumption spending and the structure of the article from levels and paragraphs.

1.Response:

We have rewritten the introduction in the amended paper to satisfy your requests. The amendments are as follows:

Household consumption is critical to stimulating economic growth. Consumption is the engine, source, and goal of social production and development. Moreover, it is a considerable driver and a significant and enduring contributor to economic growth. The study of consumer spending is a critical economic issue. The proportion of consumer spending in total expenditure exceeds about 90% of the GNI in developing countries but falls to about 60% in wealthy countries (Almosabbeh, 2020). Many economists have studied household consumption spending as one of the key determinants of a nation’s well-being (Duesenberry, 1949; Friedman, 1957; Keynes, 1936). The long-term gap between urban and rural areas in Taiwan has resulted in a meager consumption rate in its counties and cities, except in Taipei, New Taipei, Taoyuan, Taichung, Tainan, and Kaohsiung. In addition, it has also resulted in the widening consumption gap between urban and rural areas, regions, and different income groups. The two factors above have seriously hindered consumption growth in Taiwan and its contribution to economic and social development.

Regional difference has always been one of the most significant concerns of geographers, economists, and governments. Regional difference is commonplace in economic development, while residents’ consumption difference directly reflects differences in economic and social development. However, in terms of the spatial dependence of household consumption spending, economic activities of household consumption in a specific geographical area do not exist independently; there is a spatial correlation with neighboring geographical areas. The distribution of household consumption spending or consumption behavior has specific spatial rules. The amount of household consumption spending in different geographical areas may be affected by local factors and neighborhood effects. However, an important premise for traditional econometrics is to assume that the study objects are independent of one another, which does not conform to the actual situation. Traditional econometric models assume that space objects are unrelated and homogeneous, and most adopt ordinary least squares (OLS) to estimate the regression model. Due to the neglect of spatial effect, the regression model generally had errors, leading to a lack of precision regarding estimated results and inferences drawn from the regression model (LeSage & Pace, 2009). Traditional econometric models have certain limitations in spatial relations and model analysis, and it is difficult to determine factors that affect household consumption spending in counties and cities.

This study adopted the research method of spatial econometrics to explore the determinants that affect the average household consumption spending in counties and cities. A comprehensive review of previous literature shows that it is relatively rare to explore the topics of factors affecting household consumption spending from the perspective of spatial effect. This study analyzed fiscal expenditures and household characteristics from the standpoint of spatial effect. First, regarding fiscal expenditures, previous studies have demonstrated a significant relationship between government expenditures and private consumption spending (Bailey, 1971; Bernardini & Peersman, 2018; Bouakez & Rebei, 2007; Evans & Karras, 1996; Ho, 2001; Samadi & Sayedi, 2012). Whether government expenditures damage or stimulate private economic activities is a key issue in Taiwan’s economy, especially for economic revitalization in counties and cities. Therefore, it is necessary to understand how fiscal expenditures from local governments affect private household consumption spending. This study focused on the impact of productive expenditures (economic development expenditures, and educational, scientific, and cultural expenditures) from government fiscal expenditures on average household consumption. Second, in terms of household characteristics, previous studies have demonstrated that different households have different economic conditions that affect household consumption spending (Lee & Mori, 2019; Levay, Vanhille, Goedeme, & Verbist, 2021; Kozyreva, Di, Nizamova, & Smirnov, 2021; Tscharaktschiew & Hirte, 2010). Therefore, we must understand how household characteristics affect household consumption spending. This study specifically focused on the impact of household characteristics on average household consumption spending, such as average disposable income per household, the average number of employed persons per household, the ratio of self-owned residences, the average living area per person, and the ratio of catering expenses to consumption spending. As previous literature seldom used spatial effect analysis, the purposes of this study are as follows: (1) to explore whether average household consumption spending in Taiwan’s counties and cities has an overall agglomeration of spatial autocorrelation as well as the agglomeration in specific counties and cities, (2) to explore the direct effect of fiscal expenditures and household characteristics in local counties and cities on average household consumption spending in local counties and cities, (3) to explore the spatial spillover effect of fiscal expenditures and household characteristics in neighboring counties and cities on average household consumption spending in local counties and cities.

In the next chapter, we will explore the literature and outline our hypothesis development. The following research methods illustrate data and samples, research variables, and the spatial weight matrix. The empirical results include narrative statistics, correlation analysis, spatial autocorrelation verification, a LISA agglomeration map, the Hausman test, spatial Durbin model analysis, and the analysis of the direct and spatial spillover effect. Finally, we will present conclusions, policy implications, research limitations, and future research directions.

  1. In the chapter 2.1. Application of the Spatial Econometric Model in Consumption Spending, the author mentioned: there are relatively few studies on the spatial measurement of average household consumption spending. What are the main research methods used in the existing literature? Besides the research method, what are the main marginal contributions of this paper.

2.Response:

We appreciate your opinions. We have rewritten Paragraph 2 of 2.1. Application of the Spatial Econometric Model in Consumption Spending in the amended paper to satisfy your requests. The amendments are as follows:

There is limited research regarding the application of spatial econometrics on average household consumption spending. The reason is that spatial econometrics, developed to solve spatial autocorrelation problems, is a branch of econometrics where research has been far less than traditional econometrics. In the past, most research still applied traditional econometrics as the primary method. Presently, the research on the application of spatial econometrics on consumption focuses on energy consumption expenditures or medical consumption expenditures. In contrast, research on household consumption spending is relatively rare. The contribution of this study is to research household consumption spending from the method of spatial econometrics to address the gap in previous research. Moreover, this study explored the factors that affect household consumption spending from fiscal expenditures and household characteristics, which is not only a topic unstudied in previous literature but also a contribution to future research.

  1. Queen contiguity was used to define spatial weights matrix. It is suggested that the authors consider the distance and economic gap between counties and cities when constructing the spatial weight matrix.

3.Response:

We appreciate your opinions. There are various rules to construct spatial weight matrices, among which adjacency and distance rules are standard. This study adopted adjacency rules and created a spatial weight matrix with queen contiguity but did not consider constructing a spatial weight matrix with distance or economic gaps. The main reason is that this study explored the determinants affecting average household consumption spending in Taiwan’s counties and cities, including the influencing factors both in local and in neighboring counties and cities. This study emphasized the influences among counties and cities and the effects of neighboring counties and cities. Therefore, it is unsuitable to adopt distance and economic gaps for the purposes of this study.

  1. What is the author's criterion for using a spatial Durbin model rather than a spatial autocorrelation or spatial lag model? Have relevant tests been carried out.

4.Response:

We appreciate your opinions. We have demonstrated why the verification results from the spatial Durbin model were used in the first paragraph of Section 4.3. Hausman Test Results in the amended paper, rather than the spatial lag or spatial error model. The amendments are as follows:  

LeSage and Pace (2009) and Elhorst (2010) pointed out that the spatial Durbin model could be simplified as the spatial lag model or the spatial error model. The model was selected through the verification of the following hypothesis. H0: =0; if the original hypothesis was rejected, the spatial Durbin model could not be simplified to the spatial lag model. Thus, it was more appropriate to choose the spatial Durbin model. H0: +=0; if the original hypothesis was rejected, the spatial Durbin model could not be simplified as the spatial error model. Thus, it was more appropriate to select the spatial Durbin model. Between the spatial lag model and the spatial Durbin model, the Wald test indicated =34.36, p<0.001, and the Likelihood-ratio test indicated =46.07, p<0.001. Therefore, it was more appropriate to select the spatial Durbin model. Between the spatial error model and the spatial Durbin model, the Wald test indicated =47.17, p<0.001, and the Likelihood-ratio test indicated =47.83, p<0.001. Therefore, it was more appropriate to select the spatial Durbin model.

  1. Except for conclusions and policy implications, there is more room to focus on the limitations and research outlook.

5.Response:

We appreciate your opinions. We have added Section 5.3. Research Limitations and 5.4. Future Research Directions in 5. Conclusion and Policy Implications. The amendments are as follows:

5.3. Research Limitations

This section presents some research limitations in this study. First, the spatial effect includes spatial dependence (autocorrelation) and spatial heterogeneity. Spatial heterogeneity mainly consists of the heterogeneity of spatial units in shape and size and the non-stationary structure of economic phenomena in space. In this study, spatial Durbin model analysis could only solve spatial autocorrelation but could not wholly solve spatial heterogeneity. Second, in terms of research samples, there are only 22 administrative regions in Taiwan. This number is relatively small compared to other countries or regions, which may lead to fewer research samples. Third, in terms of the research period, this study examined data from 2000 to 2020. It is difficult to establish and collect panel data. If some variables are not gathered in the government database, or the data in some counties and cities are missing, it will lead to difficulties in data collection and analysis. Furthermore, the timing of information release from the Taiwan government also caused some limitations. The government has not released some variables in the statistics of counties and cities from 2021. As this study could not access the data in 2021, it took 2000 to 2020 as the research period.

5.4. Future Research Directions

This study suggests several future research directions. First, due to the spatial correlation or spatial heterogeneity among counties and cities, the impact of fiscal expenditures and household characteristics on each county and city is different. This study adopted the spatial Durbin model to empirically verify the effect of fiscal expenses and household characteristics on average household consumption spending. However, we suggest other variables are worth discussing, such as local fiscal expenditure variables. In terms of the division of expenditure structure, the expenditure structure in Taiwan's local governments can be divided into productive expenditures (educational, scientific, and cultural expenditures, and economic development expenditures), non-productive expenditures (general government affairs expenditures, social welfare expenditures, and retirement pension expenditures) and other expenditures (community development and environmental protection expenditures, police administration expenditures, debt expenditures, and assistance and auxiliary expenditures). It is suggested that subsequent researchers may find suitable variables from the literature or other theories for modeling and analysis. Second, it is recommended that subsequent research extends this topic to include spatial heterogeneity for the analysis with other, more complex, spatial econometric models. To solve the issue of spatial heterogeneity, a geographically weighted regression model (GWR) can be adopted for processing (Brunsdon, Fotheringham, & Charlton, 1996).

Reference

Brunsdon, C., Fotheringham, A.S., & Charlton, M., (1996). Geographically weighted regression: a method for exploring spatial nonstationarity. Geographical Analysis, 28, 281–298.